# Characterization and individual-level prediction of cognitive state in the first year after 'mild' stroke

Juan Pablo Saa[1,2]*, Tamara Tse[1], Gerald Choon-Huat Koh[3], Philip Yap[4], Carolyn M. Baum[5], David E. Uribe-Rivera[6], Saras M. Windecker[7], Henry Ma[8], Stephen M. Davis[9], Geoffrey A. Donnan[9], Leeanne M. Carey[1,2,10]

1 Occupational Therapy, School of Allied Health, Human Services and Sport, College of Science Health and Engineering, La Trobe University, Melbourne, Australia, 2 Neurorehabilitation and Recovery, The Florey Institute of Neuroscience and Mental Health, University of Melbourne, Melbourne, Australia, 3 Saw-Swee Hock School of Public Health, National University of Singapore, Singapore, Singapore, 4 Geriatric Medicine, Khoo Teck Puat Hospital, Singapore, Singapore, 5 School of Public Health, Washington University School of Medicine, Saint Louis, MO, United States of America, 6 Commonwealth Scientific and Industrial Research Organisation (CSIRO) of Australia, Brisbane, Queensland, Australia, 7 Telethon Kids institute, Perth, Australia, 8 Department of Medicine, Monash Health, Monash University, Clayton, Australia, 9 Departments of Medicine and Neurology, Melbourne Brain Centre, Royal Melbourne Hospital, University of Melbourne, Melbourne, Australia, 10 Care Economy Research Institute, La Trobe University, Bundoora, Australia

* saajp@outlook.com

**Data Availability Statement:** At the moment, there are still ongoing studies using the START data for predicting cognitive outcomes. Therefore, there are

## Abstract

### Background

Mild stroke affects more than half the stroke population, yet there is limited evidence characterizing cognition over time in this population, especially with predictive approaches applicable at the individual-level. We aimed to identify patterns of recovery and the best combination of demographic, clinical, and lifestyle factors predicting individual-level cognitive state at 3- and 12-months after mild stroke.

### Methods

In this prospective cohort study, the Montreal Cognitive Assessment (MoCA) was administered at 3–7 days, 3- and 12-months post-stroke. Raw changes in MoCA and impairment rates (defined as MoCA<24 points) were compared between assessment time-points. Trajectory clusters were identified using variations of ≥1 point in MoCA scores. To further compare clusters, additional assessments administered at 3- and 12-months were included. Gamma and Quantile mixed-effects regression were used to predict individual MoCA scores over time, using baseline clinical and demographic variables. Model predictions were fitted for each stroke survivor and evaluated using model cross-validation to identify the overall best predictors of cognitive recovery.

### Results

Participants' (n = 119) MoCA scores improved from baseline to 3-months (*p*<0.001); and decreased from 3- to 12-months post-stroke (*p* = 0.010). Cognitive impairment rates

restrictions on sharing a de-identified data set. The primary contact information for a data access is Prof. Leeanne Carey L.Carey@latrobe.edu.au. The best contact outside our core research team for data access queries related to the START cohort study is our hospital ethics department: ethics@austin.org.au.

**Funding:** START was supported by a Flagship Collaboration Fund through the Preventative Health Flagship and Commonwealth Scientific and Industrial Research Organisation (CSIRO) of Australia. Write-up was supported by the James S. McDonnell Foundation (grant #220020413); the National Health and Medical Research Council (NHMRC) of Australia (GNT1077898; GNT1153236; GNT2004443); and La Trobe University (La Trobe Full-Fee Research, LTUFFR, and La Trobe Postgraduate Research, LTUPR scholarships).

**Competing interests:** The authors have declared that no competing interests exist.

decreased significantly from baseline to 3-months ($p<0.001$), but not between 3- and 12-months ($p = 0.168$). Nine distinct trajectory clusters were identified. Clinical characteristics between clusters at each time-point varied in cognitive outcomes but not in clinical and/or activity participation outcomes. Cognitive performance at 3- and 12-months was best predicted by younger age, higher physical activity levels, and left-hemisphere lesion side.

## Conclusion

More than half of mild-stroke survivors are at risk of cognitive decline one year after stroke, even when preceded by a significantly improving pattern in the first 3-months of recovery. Physical activity was the only modifiable factor independently associated with cognitive recovery. Individual-level prediction methods may inform the timing and personalized application of future interventions to maximize cognitive recovery post-stroke.

## Introduction

Mild stroke affects more than half the stroke population [1]. Although there is currently no internationally recognized consensus on a measurement tool and cutoff score [2], it is common for clinicians and researchers to identify mild stroke patients using a score from 5–8 points on the National Institutes of Health Stroke Scale (NIHSS) [3, 4].

Cognitive changes are common after mild stroke, yet often overlooked and under-diagnosed [5]. Mild stroke is associated with major difficulties in common but complex activities (e.g. work, driving) that require the use of cognitive skills [6]. The Montreal Cognitive Assessment (MoCA) is a valid and reliable tool used to evaluate post-stroke cognition [7]. Studies using MoCA frequently describe associations between cognition and clinical characteristics (e.g. risk factors, comorbidities) or routine laboratory blood work to guide interventions [8–10]. However, the evidence supporting those relationships is frequently limited in their interpretation. These limitations include simplification of skewed data into impaired/non-impaired groups; and/or reduction of longitudinal outcomes into cross-sectional sub-analyses (e.g.[11, 12]).

Recent systematic evidence has provided an overall description of the quantitative changes in cognition after stroke in the short and long term, both in the absence and presence of interventions other than usual care [13]. We have also started to unveil ways in which post-stroke disability can be predicted at the individual level in mild stroke, using machine learning algorithms [14]. Despite these advances, similar machine learning approaches for cognition in mild stroke patients are still lacking.

A range of factors have potential to influence cognition and its trajectory post-stroke. These include: demographic, such as age, education and gender [15]; clinical, including initial severity of cognitive impairment, stroke severity [16]; neurological, including lesion side and volume [17]; interventions such as tissue-plasminogen activator (i.e. tPA or alteplase), anticoagulant or antiplatelet medications and their association with hemorrhagic transformation [18]; modifiable risk factors such as diet and lifestyle [19]; and blood-based biomarkers (e.g. cholesterol, vitamin B12, vitamin D, and inflammatory biomarkers) [10, 20, 21]. In the present study we focus on three main predictor categories: demographic characteristics, clinical characteristics, and lifestyle factors, given the consistent evidence found for these variables, the available data, and the potential to identify modifiable factors in this cohort of mild stroke survivors.

To improve the characterization and precision of currently utilized predictive methods of cognition after stroke, we analyzed the raw scores of the multi-factor cognitive assessment MoCA. Our aims were: (i) to characterize cognitive trajectory pathways according to patterns of change in MoCA scores, and (ii) to find the best overall combination of available baseline variables (i.e. evaluated at 3–7 days post mild stroke) predicting future cognitive state at 3 and 12 months at the individual-patient level.

## Methods

### Study design

The STroke imAging pRevention and treatment (START) study was a multisite investigation including two sub-studies: the START-EXtending the time for Thrombolysis in Emergency Neurological Deficits (EXTEND) randomized placebo-controlled trial [22]; and the START-Prediction and Prevention to Achieve Optimal Recovery Endpoints after stroke (PrePARE) cohort study [23]. Participants were recruited consecutively from June 10th, 2010 until July 4th, 2014 for the PrePARE cohort; and until June 2018 for the EXTEND trial [24]. START was approved by the Australia/New Zealand (NZ) Clinical Trials registry www.anzctr.org.au (ID# ACTRN12610000987066). Central ethical approval for this study was obtained from Melbourne Health Human Research Ethics Committee (2009.079) and Austin Health Human Research Ethics Committee (H2010/03588). Consent was obtained in written form directly from the patient, family member or legally responsible other.

The final results of the EXTEND trial were published in 2019 [24]. The current investigation involves analysis of cognition in both the START-EXTEND and START-PrePARE cohorts over the first year post-stroke using baseline demographic, clinical, and lifestyle factors as predictors of cognition at 3- and 12-months post-stroke.

### Participants

We included participants aged ≥18 years, English-speaking, with a diagnosis of mild ischemic stroke (NIHSS ≤8 points based on previously used clinical categorization of stroke severity) [4], no prior disability (modified Rankin Scale, mRS ≤2 points), and able to undertake a baseline cognitive assessment 3–7 days post-stroke.

### Assessment

Evaluations were carried out at baseline, 3- and 12-months post-stroke. Assessments were conducted in-person by trained, blinded evaluators, in a clinic testing room, or at the participant's home.

The primary outcome was general cognitive functioning, as evaluated by the MoCA [7]. Sensitivity and specificity for this assessment are available for Australia/New Zealand stroke patients, with <24 points indicating cognitive impairment [25].

### Baseline assessment

Clinical and demographic variables recorded at baseline (Table 1) and used in our predictive analysis included: age, sex, ethnicity, marital status, prior medical conditions, prior level of physical activity (Rapid Assessment of Physical Activity, RAPA) [26]; depression (Montgomery-Åsberg Depression Rating Scale, MADRS) [27]; stroke severity (National Institutes of Health Stroke Scale, NIHSS); and putative risk factors (smoking, body mass index, and blood pressure) selected based on documented associations of these variables with cognitive recovery [28, 29].

**Table 1. Baseline characteristics of sample.**

| Outcome at 3–7 days post-stroke | Levels (categorical levels and Eligible \| Included \| Excluded sample) | Eligible participants (n = 144) | Included MoCA complete (n = 119) | Excluded MoCA incomplete (n = 25) | Estimate (95% CI) | p-value* |
|---|---|---|---|---|---|---|
| Age (years) | | 66.45 (16.92) | 67.8 (15.95) | 63.8 (15.4) | 3.9 (-1.5–9.5) | 0.186 |
| Sex | Female | 47 (32.64%) | 37 (31.09%) | 10 (40%) | 0.68 (0.26–1.86) | 0.482 |
| | Male | 97 (67.36%) | 82 (68.91%) | 15 (60%) | | |
| Ethnicity | Australian/NZ | 88 (61.11%) | 72 (60.5%) | 16 (64%) | 0.86 (0.31–2.28) | 0.824 |
| | Other | 56 (38.89%) | 47 (39.5%) | 9 (36%) | | |
| Education | Primary | 19 (13.19%) | 14 (11.76%) | 5 (20%) | 0.55 (0.16–2.18) | 0.333 |
| | Secondary or more | 122 (84.72%) | 102 (85.71%) | 20 (80%) | | |
| Marital status | Married | 97 (67.36%) | 80 (67.23%) | 17 (68%) | 0.97 (0.33–2.61) | 1 |
| | Not married | 47 (32.64%) | 39 (32.77%) | 8 (32%) | | |
| Disability (mRS) | No disab | 124 (86.11%) | 102 (85.71%) | 22 (88%) | 0.82 (0.14–3.21) | 1 |
| | Some disab | 20 (13.89%) | 17 (14.29%) | 3 (12%) | | |
| Previous stroke | No | 125 (86.81%) | 106 (89.08%) | 19 (76%) | 2.55 (0.71–8.36) | 0.102 |
| | Yes | 19 (13.19%) | 13 (10.92%) | 6 (24%) | | |
| Previous TIA | No | 113 (78.47%) | 96 (80.67%) | 17 (68%) | 1.95 (0.65–5.52) | 0.184 |
| | Yes | 31 (21.53%) | 23 (19.33%) | 8 (32%) | | |
| Hypertension | No | 66 (45.83%) | 56 (47.06%) | 10 (40%) | 1.33 (0.51–3.6) | 0.66 |
| | Yes | 78 (54.17%) | 63 (52.94%) | 15 (60%) | | |
| Diabetes | No | 124 (86.11%) | 102 (85.71%) | 22 (88%) | 0.82 (0.14–3.21) | 1 |
| | Yes | 20 (13.89%) | 17 (14.29%) | 3 (12%) | | |
| Ischemic heart disease | No | 116 (80.56%) | 97 (81.51%) | 19 (76%) | 1.39 (0.41–4.19) | 0.58 |
| | Yes | 28 (19.44%) | 22 (18.49%) | 6 (24%) | | |
| Atrial fibrillation | No | 120 (83.33%) | 98 (82.35%) | 22 (88%) | 0.64 (0.11–2.43) | 0.768 |
| | Yes | 24 (16.67%) | 21 (17.65%) | 3 (12%) | | |
| Ever smoker | No | 40 (27.78%) | 32 (26.89%) | 8 (32%) | 0.78 (0.29–2.31) | 0.627 |
| | Yes | 104 (72.22%) | 87 (73.11%) | 17 (68%) | | |
| Lesion side | Left | 38 (26.39%) | 35 (29.41%) | 3 (12%) | 2.31 (0.48–14.96) | 0.318 |
| | Right | 42 (29.17%) | 35 (29.41%) | 7 (28%) | | |
| NIHSS (stroke severity) | 144 \| 119 \| 25 | 2 (3) | 2 (3) | 1 (3) | 0 (-1-1) | 0.88 |
| MADRS (depression) | 140 \| 116 \| 24 | 4 (8) | 4 (7) | 7 (15.25) | -1 (-5-1) | 0.172 |
| MoCA (cognition) | 144 \| 119 \| 25 | 25 (5) | 25 (6) | 24 (8) | 1 (-1-3) | 0.172 |
| Height | 132 \| 108 \| 24 | 170 (18) | 170 (16) | 168 (18.25) | 0 (-5-7) | 0.764 |
| Weight | 135 \| 111 \| 24 | 78 (21) | 78 (20) | 77.5 (26.25) | 0 (-10-9) | 0.916 |
| BMI | 127 \| 104 \| 23 | 26.93 (5.24) | 26.91 (4.73) | 28.7 (8.15) | -0.58 (-3.11-1.79) | 0.628 |
| Systolic BP | 144 \| 119 \| 25 | 138 (26.5) | 140 (25.5) | 131 (29) | 1 (-8-11) | 0.667 |
| Diastolic BP | 144 \| 119 \| 25 | 77 (14) | 77 (13) | 80 (16) | -1 (-7-5) | 0.562 |

(*Continued*)

**Table 1.** (Continued)

| Outcome at 3–7 days post-stroke | Levels (categorical levels and Eligible \| Included \| Excluded sample) | Eligible participants (n = 144) | Included MoCA complete (n = 119) | Excluded MoCA incomplete (n = 25) | Estimate (95% CI) | p-value* |
|---|---|---|---|---|---|---|
| RAPA aerobic | 144 \| 119 \| 25 | 4 (3) | 4 (3) | 4 (3) | 0 (-1-1) | 0.955 |
| RAPA strength/ flexibility | 144 \| 119 \| 25 | 0 (1) | 0 (1) | 0 (0) | 0 (0–0) | 0.086 |
| Charlson Cmb. Index | 144 \| 119 \| 25 | 3 (2) | 3 (2) | 3 (0) | 0 (0–0) | 0.162 |

*p-values from Fisher's exact and Wilcoxon rank-sum tests.

**BP** = Blood pressure; **BMI** = Body mass index; **Cmb** = Comorbidity; **MADRS** = Montgomery-Åsberg Depression Rating Scale; **MoCA** = Montreal Cognitive Assessment; **mRS** = modified Rankin Scale; **NIHSS** = National Institutes of Health Stroke Scale; **RAPA** = Rapid Assessment of Physical Activity; **TIA** = Transient Ischemic Attack

### Additional clinical, imaging, and cognitive assessments

Additional variables monitored at each time-point (i.e. baseline, 3, and 12-months post-stroke) included depression (MADRS), stroke severity (NIHSS), and blood pressure.

A subgroup of stroke participants (n = 98) enrolled in the START-PrePARE study underwent advanced brain-imaging to confirm lesion side at 3 months [23]. This subgroup also underwent additional assessments at 3- and 12-months to monitor general cognition (Mini-Mental State Examination, MMSE [30]); executive function (Stroop test [31], Trail Making Test, part B [32], and Raven's Progressive Matrices, RPM [33]); memory (Digit Span Forward and Backward [34]); disability (Barthel Index, BI [35]; and modified Rankin Scale, mRS [36]); participation (Activity Card Sort, ACS, Australian Version) [37]; work and social adjustment (Work and Social Adjustment Scale, WSAS) [38]; and stroke impact (Stroke Impact Scale, SIS [39]).

### Risk of bias

Assessors were blinded to study outcomes [22, 23]. All participants were tested at their recruiting hospital or home, by trained health professionals. Associations between MoCA and explanatory variables (predictors) at baseline were adjusted statistically to control for possible confounders/effect modifiers in the model validation process, as detailed in our statistical analysis. In addition, we conducted comparative post-hoc analyses to study potential effect modifiers from the use of medication such as antiplatelet, tPA, or anticoagulant during the course of the study.

### Statistical analysis

Baseline characteristics between included and excluded participants were summarized and compared using Wilcoxon rank-sum or Fisher's exact tests, to compare continuous and categorical variables, respectively.

### Characterizing cognitive recovery pathways

Changes in cognitive impairment rates between time-points were compared with the McNemar's $\chi^2$ test [40]. Differences in MoCA scores between time-points were compared using the Wilcoxon-Pratt signed-rank test [41]. This analysis was conducted both for the whole sample, and subsequently, for survivors displaying the same cognitive trajectory pattern (i.e. belonging to the same trajectory cluster, as explained below).

Trajectory clusters were created using MoCA score variations of ≥1 point between assessments. The resulting clusters were then described and compared to look for differences in

demographic and clinical profiles at each time-point. We also looked for participants with a clinically significant improvement ($\geq$2 points) in MoCA scores between assessments, as defined by previous evidence [42].

### Identifying predictors of cognition

A visual guide explaining the variable selection process we completed as part of our exploratory analyses is depicted in Fig 1. The selection of potential variables for our final model validation involved an exploratory phase in which we employed unadjusted and adjusted analyses, using groups of variables from three predictor categories (i.e. demographic characteristics, clinical characteristics, and lifestyle factors). The skewed nature of the primary outcome (i.e. overall MoCA score), led us in the first instance to select binary and quantile regressions, in line with previous literature [11, 12]. For these models, each baseline variable from Table 1 was studied, one at a time, to explore their predictive association with cognition at each time-point (i.e. baseline, 3 and 12-months).

Two additional model formulations were used in this exploratory phase to account for the distribution of our outcome data and the repeated measurements: linear-quantile mixed-models (LQMM) [43] and Gamma-distributed mixed-linear regression models (henceforth called "mixed Gamma models") [44, 45]. These two modeling approaches take into account both the skewed nature of the overall MoCA scores, as well as the follow-up (i.e. longitudinal) measurements recorded for each participant. For each model formulation, we conducted additional exploratory, unadjusted, bivariate analyses, using overall MoCA scores (all 3 time-points at once) as the outcome (i.e. dependent) variable; and one predictor (i.e. independent) variable

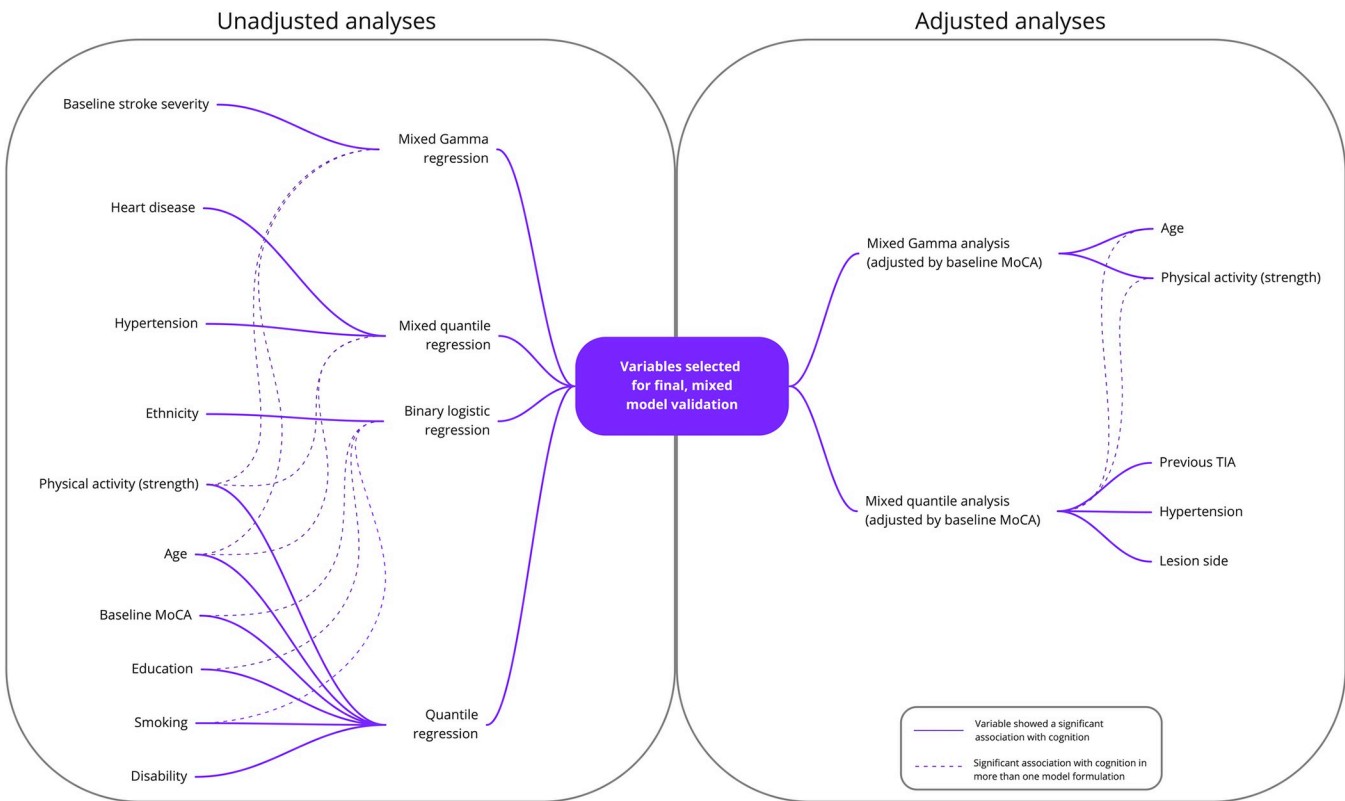

**Fig 1. Exploratory unadjusted and adjusted regression analyses used for final model variable selection** (N = 119).

from Table 1, introduced one at a time. The same models were then re-run adjusting for baseline MoCA score to predict subsequent MoCA scores at 3 and 12-months post-stroke.

## Identifying the 'best' combinations of predictive variables for cognition, based on individual-patient prediction

After finalizing our exploratory analyses, a set of candidate multivariate formulae were constructed, combining the baseline variables that showed a significant association with MoCA in the unadjusted (i.e. bivariate), and adjusted analyses (S1 Table). These multivariate formulae were limited to a maximum of three predictors (i.e. one categorical, plus two continuous), adjusting for baseline MoCA scores each time. Pseudo-replication was controlled for by including each participant as a random intercept (i.e. random effect) in all models. Each model was fitted using the 'k-fold' cross-validation, also known as 'model training' method [46], which consists of fitting a model iteratively after removing one participant in each iteration. The resulting model is then tested on the held-out individual to predict their scores. This way, each individual has the opportunity to be the 'test' data.

After predicting scores for each individual person with all combinations of candidate baseline variables, we compared predicted and observed scores for each individual. Overall best predictive performance was arrived at by calculating a Pearson's correlation between predicted and observed scores for each individual in each model.

## Power calculations

Power in the original study protocol was based on a model with 200 participants, and seven predictors at medium predictive capacity ($R^2 = 0.5$) [23]. Given our smaller sample size, in our pre-analysis phase, we simulated a GLMM model, using a Gamma distribution for a maximum of five predictors, with 0.8 power, and alpha set at 0.05. Subsequently in our main analysis, we used a conservative approach and limited the number of predictors to a maximum of four variables (three baseline predictors, all adjusted by baseline MoCA).

## Model external validation

Our modelling approach was externally validated on an independent mild stroke cohort from Singapore evaluated at admission (i.e. within the first 24 hours after the presentation of stroke symptoms), 3- and 12-months post-stroke [47]. Variables that were common across the two cohorts were mapped and then used as baseline predictors. Models in the Singapore cohort were tested utilizing the same 'k-fold' validation and model selection methods described above. Comparisons between these two cohorts, and external validation results can be found in our (S1 and S4 Tables; and S1 and S2 Figs).

## Software and predictive algorithms

All analyses and figures were completed using R v.3.5 [48]. The *LQMM* [43] and *LME4* [45] packages were used to fit mixed quantile, and Gamma models, respectively. Prediction of individual MoCA scores in *LQMM* was achieved by adapting the package's original source code to predict individual scores (LQMM's 'predict' function only predicts outcomes for the full sample and not for one individual, see our R-code for full details on this step). The full descriptive and predictive analysis plan and corresponding R-code for this study are available on https://github.com/jpsaa/saa_cog_recovery_2023.

# Results

## Cohort characteristics

One-hundred and forty-four participants met entry criteria for this study; twenty-five (17%) were lost to follow-up. Of those that were lost to follow-up (LTFU), 10 and 15 were not evaluated at 3- and 12-months, respectively, leaving a total of 119 individuals (83% of the enrolled sample) with complete data for analysis. Comparisons between included and excluded individuals (Table 1) did not reveal differences in any demographic or clinical characteristics at baseline.

## Cognitive recovery pathways

**Cognitive impairment in the first year post-stroke.** There were 45/119 (38%) participants with cognitive impairment (MoCA <24) at baseline (Fig 2), 20/119 (17%) at 3-months, and 27/119 (23%) at 12-months. Change in impairment rates from baseline to 3-months was significant (McNemar's $\chi^2$ (1) = 15.568; $p$<0.001) with 31/45 (69%) impaired participants progressing to no impairment (score ≥24 points), and 6/74 (8%) non-impaired participants developing cognitive impairment at 3-months. From 3- to 12-months, 6/20 (30%) impaired participants progressed to no impairment; while 13/99 (13%) developed cognitive impairment (McNemar's $\chi^2$ (1) = 1.89; $p$ = 0.168). Overall, from baseline to 12-months, 25/45 (56%) participants with cognitive impairment progressed to no impairment; while 7/74 (9%) unimpaired participants developed impairment at 12-months (McNemar's $\chi^2$ (1) = 9.031; $p$ = 0.002). Clinical improvement in cognition at 3-months was observed in 39/45 (87%) of participants with impairment at baseline. Clinical improvement in cognition at 12-months was achieved by 36/45 (80%) participants relative to 3-months. Conversely, there were 15/74 (20%) and 20/74 (27%) participants without impairment at baseline who declined by 2 points or more at 3- and 12-months, respectively.

## Domain-specific cognitive performance over time

A median overall increase of 2 points, interquartile range (IQR = 3.5) was observed from baseline to 3-months ($p$<0.001, Table 2); and a median decline of 1 point (IQR = 3) was observed from 3 to 12-months ($p$ = 0.001). Executive/visuospatial functions, attention, language, abstraction, and delayed recall improved significantly at 3-months (Table 2). A significant improvement in overall MoCA score, executive/visuospatial and delayed recall domains was observed from baseline to 12-months.

## Cluster pathways of cognitive recovery

Nine cognitive trajectory clusters were identified based on overall MoCA changes of ≥1 point between assessments (Fig 3). The most common trajectory was given by cluster 3 (i.e. those who improved in overall MoCA from baseline to 3-months, but then declined from 3- to 12-months: "Improved-declined" cluster; 46/119 participants; 39%). Participants in this cluster scored a median (IQR) of 23.5 (6) points in the MoCA at baseline; 28 (3) points at 3-months; and 24.5 (6) at 12-months (Table 2), displaying the same significant variations in domain-specific scores as the full sample, except for executive function, which increased significantly between baseline from 4 (3) to 5 (1) points and 3-months ($p$<0.001), but then declined significantly from 5 (1) to 4 (2) points at 12-months ($p$<0.001), with an overall, non-significant change from baseline to 12-months ($p$ = 0.715).

The second most common cluster were those who showed a consistent increase in MoCA scores between evaluations (cluster 1; 22/119 participants; 18%). These participants started

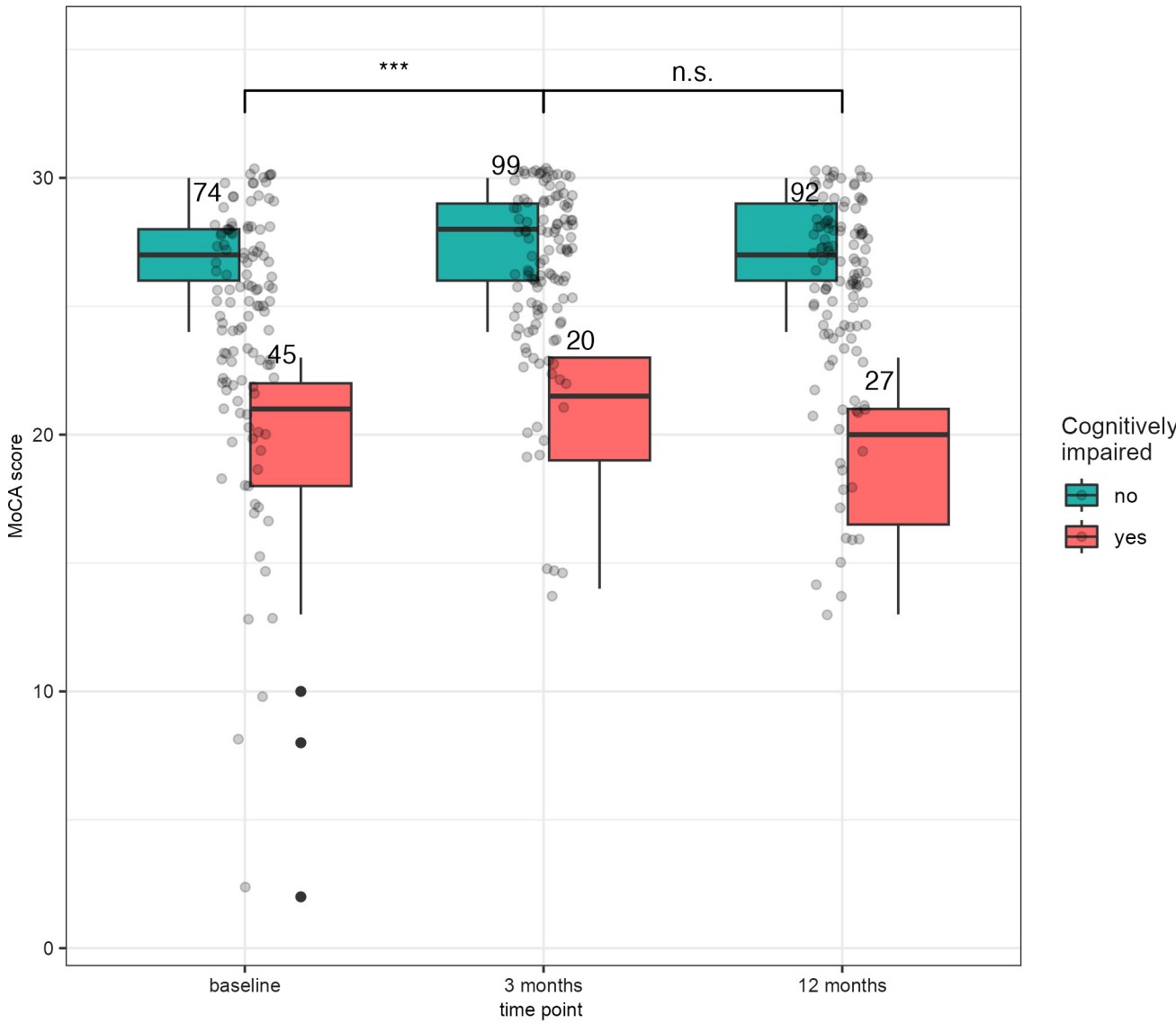

**Fig 2. Cognitively impaired and unimpaired stroke participants based on MoCA scores <24 points (N = 119).**

with an average MoCA of 22 (3) points at baseline, and then improved to 25 (1.75), and 27.5 (2.75) points at 3- and 12-months, respectively.

## Clinical profiles of trajectory clusters

Post-hoc comparative analyses of the two most common clusters (3 and 1) at each time-point (S3 Table), revealed no significant differences in MoCA scores at baseline (Wilcoxon-Mann-Whitney Z-value = -1.51; $p$ = 0.132); but significant differences in subsequent MoCA assessments at 3-months (Wilcoxon-Mann-Whitney Z-value = -2.89; $p$ = 0.003), and 12-months (Wilcoxon-Mann-Whitney Z-value = 3.28; $p$ = 0.001), as expected. Comparison of secondary evaluations (blood pressure, depression, disability, physical activity, work and social

**Table 2. Montreal Cognitive Assessment (MoCA) scores at baseline, 3- and 12-months post-stroke for the full sample (n = 119).**

| Item (range) | Baseline Median (IQR) | 3-months Median (IQR) | z-value (95% CI) | Change 1 (p-value) | 12-months Median (IQR) | z-value (95% CI) | Change 2 (p-value) | Overall z-value (95% CI) | Overall change (p-value) |
|---|---|---|---|---|---|---|---|---|---|
| Total score (0–31 points) | 25 (6) | 27 (4) | -2 (-2.5, -1.5) | < .001 | 26 (4) | 0.5 (0, 1) | 0.01 | -1 (-2, -0.5) | 0.001 |
| Exec/Visuosp (0–5 points) | 4 (2) | 4 (1) | -0.5 (-1, -0.5) | < .001 | 5 (2) | 0 (0, 0.5) | 0.066 | -0.5 (-0.5, 0) | 0.022 |
| Naming (0–3 points) | 3 (0) | 3 (0) | 0 (0, 0) | 0.522 | 3 (0) | 0 (0, 0) | 1 | 0 (0, 0) | 0.522 |
| Attention (0–6 points) | 9 (2) | 10 (1) | -0.5 (-0.5, 0) | < .001 | 9 (2) | 0 (0, 0.5) | 0.01 | 0 (-0.5, 0) | 0.073 |
| Language (0–3 points) | 6 (2) | 6 (1) | 0 (-0.5, 0) | 0.087 | 5 (2) | 0 (0, 0.5) | 0.183 | 0 (0, 0) | 0.743 |
| Abstraction (0–2 points) | 2 (1) | 2 (0) | 0 (0, 0) | **0.008** | 2 (0) | 0 (0, 0) | 0.247 | 0 (0, 0) | 0.247 |
| Delayed recall (0–5 points) | 3 (2) | 4 (2) | -0.5 (-1, -0.5) | < .001 | 4 (2) | 0.5 (0, 0.5) | 0.009 | -0.5 (-0.5, 0) | 0.009 |
| Orientation (0–6 points) | 6 (0) | 6 (0) | 0 (0, 0) | 0.727 | 6 (0) | 0 (0, 0) | 0.727 | 0 (0, 0) | 0.727 |
| **MoCA scores at baseline, 3- and 12-months post-stroke for cluster 3 "improved-declined" (n = 46)** | | | | | | | | | |
| Total score (0–31 points) | 24 (6) | 28 (3) | -4 (-4.5, -3) | < .001 | 25 (6) | 2.5 (2, 4) | < .001 | -1 (-2, 0) | 0.047 |
| Exec/Visuosp (0–5 points) | 4 (3) | 5 (1) | -1 (-1.5, -0.5) | < .001 | 4 (2) | 0.5 (0.5, 1) | 0.001 | 0 (-0.5, 0.5) | 0.875 |
| Naming (0–3 points) | 3 (0) | 3 (0) | 0 (0, 0) | 0.375 | 3 (0) | 0 (-Inf, 0) | 0.375 | 0 (0, 0) | 0.537 |
| Attention (0–6 points) | 10 (2) | 10 (1) | -0.5 (-1, 0) | 0.06 | 9 (1.5) | 0.5 (0, 0.5) | 0.166 | -0.5 (-1, 0) | 0.215 |
| Language (0–3 points) | 5 (2) | 6 (1) | -0.5 (-1, 0) | < .001 | 5 (2) | 0.5 (0.5, 1) | < .001 | 0 (-0.5, 0.5) | 0.758 |
| Abstraction (0–2 points) | 2 (1) | 2 (0) | 0 (-0.5, 0) | **0.048** | 2 (1) | 0 (0, 0) | 0.414 | 0 (0, 0) | 0.415 |
| Delayed recall (0–5 points) | 2 (1.5) | 4 (2) | -1.5 (-1.5, -1) | < .001 | 3 (2) | 0.5 (0.5, 1) | < .001 | -0.5 (-1, 0) | 0.007 |
| Orientation (0–6 points) | 6 (0) | 6 (0) | 0 (0, 0) | 0.094 | 6 (1) | 0.25 (0, 0.5) | 0.0 | 0 (0, 0.5) | 0.492 |

All p-values were obtained from Asymptotic Wilcoxon-Pratt signed-rank tests and adjusted for multiple comparisons using the Benjamini and Hochberg [66] method.
**Change 1** = comparison of MoCA at baseline and 3-months post-stroke; **Change 2** = comparison of MoCA at 3- and 12-months post-stroke; **Overall change** = change from baseline to 12-months.

**Exec/visuosp** = executive and visuospatial function; **IQR** = Interquartile range

adjustment, and impact of stroke) revealed no significant differences at each time-point between these clusters, except for activity participation (ACS) at 3-months. No significant differences were found in clinical and complementary cognitive outcomes at 3- or 12-months between clusters 3 and 1.

Comparison of clusters with opposite trajectories (cluster 1 and 9), revealed significant differences in MoCA scores at baseline and 12-months; and in Stroop test and blood pressure at 3-months (S3 Table).

## Unadjusted bivariate models

All factors investigated that had a significant association with cognition can be found in S1 Table. At baseline, better cognition was significantly associated with higher educational level

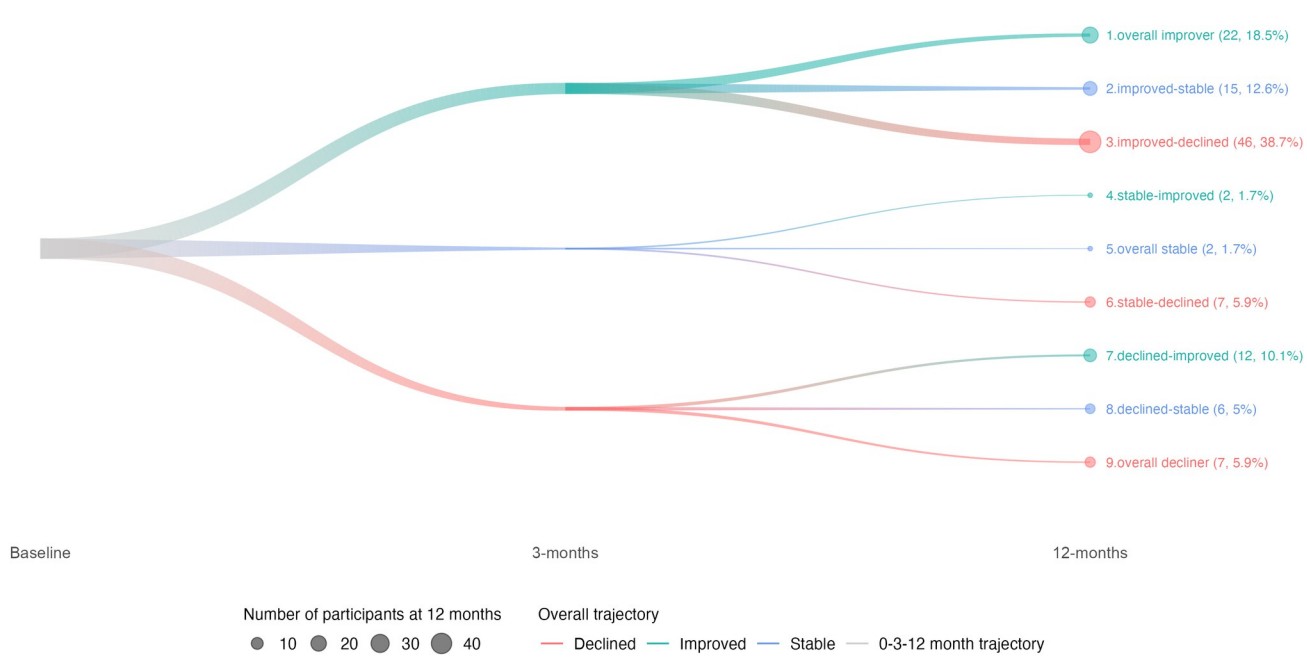

**Fig 3. Cognitive trajectory clusters based on MoCA score changes of ≤1 point at each time point (N = 119).**

(high school or more). At 3-months, better cognition was associated with higher baseline MoCA, educational level, non-smoking status, higher physical activity level (RAPA strength-flexibility score), and ethnicity (Australia/NZ). At 12-months; better cognition was associated with higher baseline MoCA, education, non-smoking status, younger age, higher physical activity levels, and the absence of premorbid disability.

## Adjusted models

Table 3 summarizes the associations between MoCA scores at 3-and 12-months and different baseline predictors, using one predictor at-a-time and adjusted by baseline MoCA. Potential baseline predictors investigated were all those listed in Table 1. The adjusted analyses presented in Table 3 revealed significant associations between MoCA at 3- and 12-months, and

**Table 3. Mixed quantile and Gamma regression models predicting Montreal Cognitive Assessment (MoCA) scores at 3- and 12-months using baseline predictors.**

| Dependent variable (MoCA) | Time-point | Baseline predictors | Estimate (95% CI)* | p-value |
|---|---|---|---|---|
| | | | **Mixed quantile regression†** | |
| MoCA median score | Longitudinal analysis | Previous TIA | -1.63 (-3.065 to -0.194) | 0.027 |
| | | Hypertension | -1.065 (-1.892 to -0.237) | 0.013 |
| | | Lesion side (right) | -0.048 (-0.077 to -0.018) | 0.002 |
| | | Age | 0.464 (0.082 to 0.846) | 0.018 |
| | | RAPA strength score | -1.63 (-3.065 to -0.194) | 0.027 |
| | | | **Mixed Gamma regression†** | |
| MoCA Gamma score | Longitudinal analysis | Age | 1.126 (0.277 to 1.975) | 0.009 |
| | | RAPA strength score | -0.899 (-1.775 to -0.023) | 0.044 |

*Estimate represents slope of linear relationships

†Models were adjusted by MoCA scores at baseline

baseline records of previous transient ischemic attack (TIA), hypertension, left-hemisphere lesion side, age, and physical activity (RAPA strength score).

## Model cross-validation

Model cross-validation revealed an almost identical correlation between predicted and observed scores for both Gamma and Quantile regression mixed-effects models (S1 Fig). The model with the best predictive accuracy in both model formulations included lesion side (left), younger age, and higher physical activity ($r = 0.7$ for mixed-quantile and 0.69 for Gamma regression).

## Impact of tPA, anticoagulant or antiplatelet medication on cognition

Lastly, we compared those participants on tPA, anticoagulant or antiplatelet medication for any reason (n = 31 participants, which includes those who were in the EXTEND RCT and received tPA treatment i.e. n = 6), versus those who did not receive any of those medications (n = 88 participants) throughout the study. Our post-hoc analyses yielded no significant differences in overall cognition between the two groups of participants at baseline, 3- or 12-months; and no significant differences in baseline variables. Please refer to S6 Table for more details.

# Discussion

Our aims were to characterize patterns of cognitive recovery over the first year post-stroke according to variations in MoCA scores; and to find the best combination of baseline variables predicting future cognitive recovery at 3- and 12-months post-stroke.

## Changes in overall and domain-specific cognition at baseline, 3- and 12-months

We described changes in overall raw scores, in addition to the changes in proportion of survivors classified using a binary categorization of cognitive state (e.g. impaired/unimpaired), using a cutoff score of <24 points, as per evidence in Australia/New Zealand cohorts [25]. When analyzing changes in raw MoCA scores, we found significant variations between all assessment time-points (i.e. from baseline to 3-months; from 3- to 12-months; and from baseline to 12-months). Our post-hoc analyses confirmed an initial improvement in overall MoCA scores explained by a significant increase in executive/visuospatial functions, attention, language, abstraction, and recall domains. The subsequent decrease was explained by poorer performance in attention and recall domains.

Binary analyses yielded significant variations in cognitive state (impaired/unimpaired) only at 3-months, and between baseline and 12-months, but not from 3- to 12-months. This finding highlights the importance of using the full range of scores (0–30 points) as a more sensitive approach than binary analyses to capture changes in cognition between assessment time-points. Further, the dichotomization of continuous variables is a well-known approach in clinical research that is not recommended as a robust approach, as it is highly likely to lead to the loss of statistical power [49, 50]. Outcome dichotomization, however, is still a common practice in stroke research [11, 12].

Overall, these findings establish that a significant decline in cognition is possible within the first year post-stroke, even when preceded by a significantly improving pattern. The importance of detecting these changes aligns with current evidence supporting the use of MoCA early after stroke and across the continuum of stroke recovery [5, 51, 52]. The MoCA is an

informative, brief, easy-to-use instrument that is known among allied-health professionals, and has shown good predictive capacity informing future disability and mortality [5, 8, 53].

## Trajectory clusters

Pathway cluster analyses revealed that the two most common cognitive trajectories were given by those survivors who improved and then declined ("improved-declined" cluster, 39%), followed by those who had a consistent improvement between assessments ("overall improvers" 18%). This finding highlights the relatively high proportion of patients–more than half our cohort–who are at risk of cognitive decline between 3- and 12-months post-stroke., as portrayed by clusters 2 ("improved-declined") and 6 ("stable-declined")(Fig 3 and S3 Fig).

Furthermore, clinical and cognitive comparison between the two larger clusters ("improved-declined" and "overall improvers") revealed that the differences between these two clusters were almost exclusively found in cognitive performance at each time-point, but not in secondary clinical evaluations potentially associated with cognition (i.e. depression scores, disability, physical activity levels, work and social adjustment and stroke impact).

Again, these findings identify cognition as a key distinguishing feature, further highlighting the value of monitoring cognitive symptoms over time.

## Model formulation–Quantile versus Gamma models

Our predictive approach using mixed Gamma vs quantile regression proved to be similarly valid and equally precise for both model formulations. Estimates of cognitive recovery were virtually identical between models, across all combinations of baseline factors we evaluated. Although our variables of interest have also been identified in previous studies [9, 11, 12, 28, 29, 42, 54], the cross-validation methodology used herein allowed us to go one step further to evaluate predictive performance for each combination of baseline variables on each stroke survivor individually. The end result was the ability to predict future scores at 3- and 12-months, using variables that are commonly evaluated at admission. A granular modelling approach that incorporates model cross-validation is an important addition to the present state of research in stroke that can help guide practitioners seeking more precise methods of prediction. This approach has the potential for future development as our algorithms have been made publicly available for researchers to modify, improve, and test. In addition, these tools are capable of modeling skewed, longitudinal outcomes, which are flexible to predict cognition, and any other continuous outcome yielding itemized or composite scores (e.g. disability, depression, pain). As new data and analysis tools become available, the use of mixed Gamma and quantile models should lead to more precise prediction algorithms with respect to popular binary and cross-sectional analyses.

Overall, our predictive findings align with previous evidence describing education, age, initial stroke severity, and physical inactivity as significant predictors of cognitive function [9, 11, 42]. Side of lesion is a relatively newly studied factor [55], that points to better sensitivity of the MoCA to detect left, more than right, hemisphere deficits [56].

## Benefit of using mixed Gamma over mixed quantile models for prediction of skewed, longitudinal outcomes

The advantage of Gamma models over LQMM is that they have been optimized to handle offset variables [45]. An offset variable, also referred to as 'known parameter', has shown to improve the accuracy of predictions in mixed models [57]. In our case, MoCA at baseline (i.e. the known parameter) could be used to scale the predictions at 3- and 12-months, instead of

using this variable to adjust each model as a fixed parameter (i.e. predictor), ultimately leading to the use of less degrees of freedom, and thus less prediction error.

## Implications for clinical practice

Overall, our predictive analyses revealed that the amount of physical activity, as assessed using the RAPA [26], was the only modifiable factor independently associated with cognition. This is a key finding of the present research that supports recent evidence underlining the positive impact of physical activity pre- and post-ischemic (and hemorrhagic) stroke on survival and overall functional recovery [58, 59]. Physical activity programs for stroke survivors have consistently been showing positive associations with cognitive recovery in systematic evidence and quality trials that have coupled these interventions with behavioral programs [29, 60, 61]. Both the present study and the accumulated evidence suggest that both primary and secondary prevention programs involving physical activity, alongside a monitoring protocol for cognition may be important not just for monitoring cognition, but also overall recovery after stroke. Furthermore, our findings provide additional detail about the positive role of physical activity on cognitive recovery, and possibly overall post-stroke functional recovery and survival.

Future studies should continue to thoroughly examine the role of physical activity in cognitive recovery and prioritize similar interventions, including a direct comparison of different types of physical activity, at different times of the subacute phase, focusing on defined windows of recovery (i.e. recovery epochs), and for varying intensities, both combined, and in isolation. The interaction (i.e. multiplicative effect) with other behavioral approaches such as self-management programs should also be considered using suitable methods of analysis that are already available from the present study and from previous systematic evidence analyzing the multiplicative effect of different interventions on cognition [13].

## Limitations

Although Gamma models have been validated widely in areas of epidemiology and biology [62], LQMM models need further development and validation for the purposes of individual prediction. As detailed in our statistical analysis, we modified LQMM's *'predict'* function to calculate scores at 3- and 12-months for a single individual. Though our modifications proved to be effective with the available data from the START cohort in Australia/NZ, and with our external validation using data from Singapore, further validation of the modified function is required, with additional datasets using MoCA and other continuous outcomes with a skewed distribution.

The limited variability in explanatory variables may also have resulted in them showing a reduced predictive value, increasing the risk of spurious variable selection. Our findings may also be masked by the clear ceiling effects of certain subtests of the MoCA such as naming (recognition of known objects) and basic orientation (time/space) [53].

Although our study did not include a comprehensive examination of vascular vulnerabilities and hemorrhagic transformation on cognition, we acknowledge the importance of monitoring these variables and including them in future studies that set out to understand the many facets of post-stroke cognitive trajectory [63, 64]. Our findings are only valid for those participants with mild stroke severity over the first year of recovery. Monitoring cognition is crucial given the current state of evidence of mild-stroke and cognitive decline in the first year post-stroke [65].

## Conclusion

Cognitive performance in mild stroke survivors improved significantly from baseline to 3-months, but then decreased significantly from 3- to 12-months post-stroke. The description

of clusters of cognitive change in the first year post-stroke has provided new insights on cognitive recovery, highlighting the importance of monitoring changes in cognition over time. MoCA scores during the first year post-stroke were consistently associated with non-modifiable factors such as age and lesion side, and with only one modifiable factors: physical activity. Future predictive studies should focus on predicting individual-level cognitive function and on finding other variables associated with cognitive recovery that are modifiable during the first year post-stroke and beyond.

## Supporting information

**S1 Table. Baseline characteristics of Australian and Singapore stroke cohorts for common variables across datasets.**
(DOCX)

**S2 Table. Changes in MoCA scores from baseline to 3- and 12-months post-stroke for group 3 "improved-declined" (n = 45) in START cohort.**
(DOCX)

**S3 Table. Exploratory, unadjusted regression analyses predicting Montreal Cognitive Assessment (MoCA) using baseline variables in START cohort (n = 119).**
(DOCX)

**S4 Table. Trajectory profiles of stroke clusters at baseline, 3- and 12-months post-stroke in START cohort, based on changes in MoCA scores.**
(DOCX)

**S5 Table. Post-hoc descriptive profile comparison of the two most common cognitive trajectory clusters in START cohort based on MoCA scores.**
(DOCX)

**S6 Table. Comparison of START participants on antiplatelet, tPA, or anticoagulant medication (n = 31) versus those not on those medications (n = 88).**
(DOCX)

**S1 Fig. Gamma versus linear quantile mixed method comparison for START cohort study.**
(TIF)

**S2 Fig. Gamma versus linear quantile mixed method comparison for Singapore cohort study.**
(TIF)

**S3 Fig. Improved-declined group (n = 46) in START cohort.** Individual trajectory of overall MoCA scores.
(TIF)

**S1 File.**
(DOCX)

## Author Contributions

**Conceptualization:** Juan Pablo Saa, Tamara Tse, Gerald Choon-Huat Koh, Philip Yap, Carolyn M. Baum, David E. Uribe-Rivera, Saras M. Windecker, Henry Ma, Stephen M. Davis, Geoffrey A. Donnan, Leeanne M. Carey.

**Data curation:** Juan Pablo Saa.

**Formal analysis:** Juan Pablo Saa, David E. Uribe-Rivera, Saras M. Windecker.

**Funding acquisition:** Juan Pablo Saa, Gerald Choon-Huat Koh, Philip Yap, Henry Ma, Geoffrey A. Donnan, Leeanne M. Carey.

**Investigation:** Juan Pablo Saa, Gerald Choon-Huat Koh, Philip Yap, Henry Ma, Stephen M. Davis, Geoffrey A. Donnan, Leeanne M. Carey.

**Methodology:** Tamara Tse, Gerald Choon-Huat Koh, Philip Yap, Carolyn M. Baum, David E. Uribe-Rivera, Saras M. Windecker, Henry Ma, Stephen M. Davis, Geoffrey A. Donnan, Leeanne M. Carey.

**Project administration:** Juan Pablo Saa, Philip Yap, Carolyn M. Baum, Henry Ma, Geoffrey A. Donnan, Leeanne M. Carey.

**Resources:** Juan Pablo Saa, Gerald Choon-Huat Koh, Henry Ma, Stephen M. Davis, Geoffrey A. Donnan, Leeanne M. Carey.

**Software:** Juan Pablo Saa, Saras M. Windecker.

**Supervision:** Juan Pablo Saa, Tamara Tse, Carolyn M. Baum, Leeanne M. Carey.

**Validation:** Juan Pablo Saa, Tamara Tse, Gerald Choon-Huat Koh, Philip Yap, Carolyn M. Baum, Saras M. Windecker, Henry Ma, Geoffrey A. Donnan, Leeanne M. Carey.

**Visualization:** Juan Pablo Saa, David E. Uribe-Rivera, Saras M. Windecker.

**Writing – original draft:** Juan Pablo Saa.

**Writing – review & editing:** Juan Pablo Saa, Tamara Tse, Gerald Choon-Huat Koh, Philip Yap, Carolyn M. Baum, David E. Uribe-Rivera, Saras M. Windecker, Henry Ma, Stephen M. Davis, Geoffrey A. Donnan, Leeanne M. Carey.

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
