## [Decision Letter · Decision Letter 0]

5 Dec 2023

PONE-D-23-28261Characterization and individual-level prediction of cognitive state in the first year after ‘mild’ strokePLOS ONE

Dear Dr. Saa,

Thank you for submitting your manuscript to PLOS ONE. After careful consideration, we feel that it has merit but does not fully meet PLOS ONE’s publication criteria as it currently stands. Therefore, we invite you to submit a revised version of the manuscript that addresses the points raised during the review process.

We look forward to receiving your revised manuscript.

Kind regards,

Burak Yulug

Academic Editor

PLOS ONE

Journal Requirements:

3. Please expand the acronym “NHMRC, CSIRO” (as indicated in your financial disclosure) so that it states the name of your funders in full.

"START was supported by a Flagship Collaboration Fund through the Preventative Health Flagship (CSIRO) of Australia. Write-up was supported by the James S. McDonnell Foundation (grant #220020413); the NHMRC of Australia (grant #1077898); and La Trobe University (LTUFFR and LTUPR scholarships)."

6. We note you have included a table to which you do not refer in the text of your manuscript. Please ensure that you refer to Table 3 in your text; if accepted, production will need this reference to link the reader to the Table.

7. Please include a copy of Table 5 which you refer to in your text on page 13.

Reviewers' comments:

Reviewer's Responses to Questions

**Comments to the Author**

1. Is the manuscript technically sound, and do the data support the conclusions?

Reviewer #1: Yes

Reviewer #2: Yes

Reviewer #3: Yes

2. Has the statistical analysis been performed appropriately and rigorously? 

Reviewer #1: Yes

Reviewer #2: Yes

Reviewer #3: Yes

3. Have the authors made all data underlying the findings in their manuscript fully available?

Reviewer #1: Yes

Reviewer #2: Yes

Reviewer #3: No

4. Is the manuscript presented in an intelligible fashion and written in standard English?

Reviewer #1: Yes

Reviewer #2: Yes

Reviewer #3: Yes

5. Review Comments to the Author

Reviewer #1: The authors well evaluated cognitive impairment in stroke patients. However, following points help to improve the general impact of the paper.

The publication does not provide any information regarding the infarct characteristics of the patients, including the infarct volume. The aforementioned characteristics can contribute to cognitive deterioration in certain individuals under observation.

The text does not provide any mention of the treatment strategies. The inquiry that arose in my thoughts was regarding the number of patients who received tPA, anticoagulant, or antiplatelet medication. Furthermore, it is well-established that the administration of statin medicine can lead to cognitive impairments. In order to mitigate the potential impact of drugs, it is imperative to take the these factors into consideration.

It is also imperative to ascertain whether the patients had hemorrhagic transformation, as this phenomenon has the potential to induce a fast increase in the National Institutes of Health Stroke Scale (NIHSS) score

and important to acknowledge the preceding instances of COVID-19 infections since COVID-19 is a subject of extensive investigation and is considered a potential cause of cognitive decline.

It is also important to include and adjust patients' blood work, such as vitamin B12, cholesterol levels, and vitamin D which are crucial parameters to lead to cognitive detoriation.

Reviewer #2: This research article offers a comprehensive analysis of cognitive recovery in patients who have experienced mild strokes. The study addresses a critical gap in understanding how cognitive abilities change over time in this specific population, aiming to identify predictive factors for individual-level cognitive states at 3- and 12-months post-stroke.

The study employs a robust methodology, including the use of the Montreal Cognitive Assessment (MoCA) to evaluate cognitive functioning and assess changes over time. The authors use trajectory clusters to categorize participants based on their cognitive recovery patterns, providing valuable insights into how different individuals may experience cognitive changes following a mild stroke.

One of the key findings of this research is the identification of physical activity as a modifiable factor independently associated with cognitive recovery. This highlights the potential role of physical activity interventions in improving cognitive outcomes for individuals recovering from mild strokes, offering important implications for clinical practice and rehabilitation programs.

The paper also discusses the limitations of the study, such as the potential for spurious variable selection due to limited variability in explanatory variables and ceiling effects in certain MoCA subtests. Additionally, the study's focus on mild stroke patients during the first year of recovery should be considered when interpreting the results.

Overall, this research article provides valuable insights into the complex trajectory of cognitive recovery after mild strokes. It underscores the importance of monitoring cognitive changes over time and offers a foundation for future studies to explore additional modifiable factors that may enhance cognitive recovery post-stroke. The findings have the potential to inform clinical practice and improve the quality of care for individuals recovering from mild strokes.

Reviewer #3: In a prospective single cohort study, the objective was to identify patterns of recovery and the best combination of baseline variables predicting individual-level cognitive state at 3- and 12-months after mild stroke. MoCA scores significantly improved from baseline to 3-months and significantly decreased from 3- to 12-months. Cognitive impairment rates decreased significantly from baseline to 3-months,but not between 3- and 12-months.

Minor revisions:

1- Line 131: Grammatical error: Fisher’s exact tests.

2- Line 171: State the alpha level which attained 80% power.

3- Explain all abbreviations at first mention.

4- Indicate if p-values were adjusted for multiple comparisons, specifically those in Table 2.

6. PLOS authors have the option to publish the peer review history of their article (what does this mean?). If published, this will include your full peer review and any attached files.

Reviewer #1: **Yes: **Dila Sayman

Reviewer #2: No

Reviewer #3: No

---

## [Author Response · Author response to Decision Letter 0]

9 May 2024

Please see file with responses attached

---

## [Decision Letter · Decision Letter 1]

5 Jul 2024

PONE-D-23-28261R1Characterization and individual-level prediction of cognitive state in the first year after ‘mild’ strokePLOS ONE

Dear Dr. Saa,

Thank you for submitting your manuscript to PLOS ONE. After careful consideration, we feel that it has merit but does not fully meet PLOS ONE’s publication criteria as it currently stands. Therefore, we invite you to submit a revised version of the manuscript that addresses the minor points raised during the review process.

We look forward to receiving your revised manuscript.

Kind regards,

Annesha Sil, Ph.D.

Staff Editor

PLOS ONE

Journal Requirements:

Reviewers' comments:

Reviewer's Responses to Questions

**Comments to the Author**

1. If the authors have adequately addressed your comments raised in a previous round of review and you feel that this manuscript is now acceptable for publication, you may indicate that here to bypass the “Comments to the Author” section, enter your conflict of interest statement in the “Confidential to Editor” section, and submit your "Accept" recommendation.

Reviewer #2: All comments have been addressed

Reviewer #3: (No Response)

2. Is the manuscript technically sound, and do the data support the conclusions?

Reviewer #2: Partly

Reviewer #3: Yes

3. Has the statistical analysis been performed appropriately and rigorously? 

Reviewer #2: Yes

Reviewer #3: Yes

4. Have the authors made all data underlying the findings in their manuscript fully available?

Reviewer #2: Yes

Reviewer #3: No

5. Is the manuscript presented in an intelligible fashion and written in standard English?

Reviewer #2: Yes

Reviewer #3: Yes

6. Review Comments to the Author

Reviewer #2: (No Response)

Reviewer #3: Minor revision:

Drop "(p)" from Line 205. In the context of statistical power, only alpha is needed. P-values are utilized when assessing the outcomes of tests.

7. PLOS authors have the option to publish the peer review history of their article (what does this mean?). If published, this will include your full peer review and any attached files.

Reviewer #2: No

Reviewer #3: No

---

## [Author Response · Author response to Decision Letter 1]

9 Jul 2024

For this second iteration of the reviewing process we did the following:

• Reviewed all of our references to make sure none of the papers we cited in our manuscript have been retracted.

• Removed “(p)” from line 205, as per the minor revision requested by reviewer #3

---

## [Editor Report · Decision Letter 2]

17 Jul 2024

Characterization and individual-level prediction of cognitive state in the first year after ‘mild’ stroke

PONE-D-23-28261R2

Dear Dr. Saa,

We’re pleased to inform you that your manuscript has been judged scientifically suitable for publication and will be formally accepted for publication once it meets all outstanding technical requirements.

Kind regards,

Laura Kelly

Division Editor

PLOS ONE
---

## [Editor Report · Acceptance letter]

24 Jul 2024

PONE-D-23-28261R2 

PLOS ONE

Dear Dr. Saa, 

I'm pleased to inform you that your manuscript has been deemed suitable for publication in PLOS ONE. Congratulations! Your manuscript is now being handed over to our production team.

Kind regards, 

on behalf of

Dr. Laura Hannah Kelly 

Staff Editor

PLOS ONE